# Integration of a (–Cu–S–)$_n$ plane in a metal–organic framework affords high electrical conductivity

Abhishek Pathak[1,2,3], Jing-Wen Shen[4], Muhammad Usman[1], Ling-Fang Wei[1], Shruti Mendiratta[1], Yu-Shin Chang[5], Batjargal Sainbileg [6,8], Chin-May Ngue[1], Ruei-San Chen [5], Michitoshi Hayashi [6,8], Tzuoo-Tsair Luo[1], Fu-Rong Chen[2], Kuei-Hsien Chen[6,7], Tien-Wen Tseng[4], Li-Chyong Chen[6,8] & Kuang-Lieh Lu[1]

Designing highly conducting metal–organic frameworks (MOFs) is currently a subject of great interest for their potential applications in diverse areas encompassing energy storage and generation. Herein, a strategic design in which a metal–sulfur plane is integrated within a MOF to achieve high electrical conductivity, is successfully demonstrated. The MOF {[Cu$_2$(6-Hmna)(6-mn)]·NH$_4$}$_n$ (**1**, 6-Hmna = 6-mercaptonicotinic acid, 6-mn = 6-mercaptonicotinate), consisting of a two dimensional (–Cu–S–)$_n$ plane, is synthesized from the reaction of Cu(NO$_3$)$_2$, and 6,6′-dithiodinicotinic acid via the in situ cleavage of an S–S bond under hydrothermal conditions. A single crystal of the MOF is found to have a low activation energy (6 meV), small bandgap (1.34 eV) and a highest electrical conductivity (10.96 S cm$^{-1}$) among MOFs for single crystal measurements. This approach provides an ideal roadmap for producing highly conductive MOFs with great potential for applications in batteries, thermoelectric, supercapacitors and related areas.

[1] Institute of Chemistry, Academia Sinica, Taipei 115, Taiwan. [2] Department of Engineering and System Science, National Tsing Hua University, Hsinchu 300, Taiwan. [3] Nano Science and Technology Program, Taiwan International Graduate Program, Academia Sinica, Taipei 115, Taiwan and National Tsing Hua University, Hsinchu 300, Taiwan. [4] Department of Chemical Engineering and Biotechnology, National Taipei University of Technology, Taipei 106, Taiwan. [5] Graduate Institute of Applied Science and Technology, National Taiwan University of Science and Technology, Taipei 106, Taiwan. [6] Center for Condensed Matter Sciences, National Taiwan University, Taipei 106, Taiwan. [7] Institute of Atomic and Molecular Sciences, Academia Sinica, Taipei 106, Taiwan. [8] Center of Atomic Initiative for New Materials, National Taiwan University, Taipei 106, Taiwan. Correspondence and requests for materials should be addressed to K.-H.C. (email: chenkh@pub.iams.sinica.edu.tw) or to T.-W.T. (email: f10403@ntut.edu.tw) or to L.-C.C. (email: chenlc@ntu.edu.tw) or to K.-L.L. (email: kllu@gate.sinica.edu.tw)

The rapid advancements in the development of electronic devices and the need for high-performance energy storage systems are leading our efforts towards the development of alternative materials that will eventually become more efficient than traditional inorganic materials[1]. In this regard, metal–organic frameworks (MOFs) are considered to be prospective candidates for use in energy storage and generation, chemical sensors and photosensitizers due to their unique features, which include self-assembly, ease of structural post-modification, versatile physical properties, and bandgap tuneability[1–5]. MOFs with an electrical conductivity of $<10^{-7}–10^{-10}\,S\,cm^{-1}$, were typically contemplated for use as insulators[6]. The low electrical conductivity of MOFs is predominantly due to the fact that carboxylate linkers are commonly used in producing multidimensional structural frameworks, where the electronegativity of the oxygen atoms of the carboxylate is so high that electrons require a higher voltage to pass through the organic linkers[3,7,8]. This results in a poor overlap between the oxygen atoms and metal $d$-orbitals, which are responsible for the low electrical conductivity of most MOFs that have been reported to date[9]. As a consequence, developing a strategy that will permit the electrical conductivity of MOFs to be systematically improved is of paramount importance.

In recent years, several methodologies have been applied to enhance the charge transport mechanism in MOFs[10–14]. Tallin, Allendorf and coworkers reported on the infiltration of conducting molecules into the pores of MOFs to produce hybrid materials with high conductivities[15]. Allendorf, Long and coworkers and D'Alessandro developed methodology that involves the use of redox-active inorganic/organic moieties in MOFs, showing that electrical conductivity could be enhanced to some extent[16–18]. Yaghi et al. reported on the preparation of a high conducting MOF (Cu$_3$(HHTP)$_2$, HHTP = 2,3,6,7,10,11-hexahydroxytriphenylene) with an electrical conductivity of 0.2 S cm$^{-1}$ in its crystal form[19]. Dinca and coworkers recently demonstrated redox-promoted electrical conductivity (1.8 S cm$^{-1}$) in {Fe$_2$(BDT)$_3$}$_n$ (H$_2$BDT = 5,5'-(1,4-phenylene)bis(1$H$-tetrazole)) in the form of a single crystal[20]. Lin, Huang and coworkers reported on the insertion of inorganic nanowires in porous MOFs to construct a semiconductor MOF[21]. Feng and coworkers also reported on the ferromagnetic semiconducting behavior of perthiolated coronene–Fe (PTC–Fe) based on a two-dimensional MOF[22]. In addition, Nishihara and coworkers demonstrated that a sulfur-containing coordination complex (Ni-bis(dithiolene)) nanosheets can exhibit conducting behavior[23]. Several groups, including Dinca and coworkers, Melot, Marinescu and coworkers, as well as He, Xu, and coworkers reported that MOFs with metal–sulfur (–M–S–)$_n$ chains showed good electrical conductivity, mainly due to the high overlap between the sulfur $p$-orbitals and the metal $d$-orbitals[24–28]. This type of bond approach, allows

charge transfer to take place via chemical bonds[26]. The (–M–S–)$_n$ chains in multidimensional MOFs have been reported to be anisotropic in nature, with the ability to facilitate charge transport in only one dimension[26].

We envisaged that expanding the (–M–S–)$_n$ chain to (–M–S–)$_n$ planes would permit charge transfer to occur along two directions, thus resulting in a MOF with an overall high electrical conductivity. Hence, we herein report on the preparation of single crystals of a Cu-based MOF with a copper–sulfur plane that shows a high electrical conductivity (10.96 S cm$^{-1}$) measured on a single crystal. As of this writing, the approach for integrating a (–M–S–)$_n$ plane in a MOF to achieve a high degree of electrical conductivity is demonstrated. This strategy represents a model for designing highly conductive MOFs and provides a stepping stone for the rapid development of highly conducting MOFs in crystalline form for applications in supercapacitors, thermoelectric, fuel cells, chemical sensing, and lithium-ion batteries[2,3,6].

## Results and discussion

**Synthesis of {[Cu$_2$(6-Hmna)(6-mn)]·NH$_4$}$_n$.** A copper–sulfur-based MOF, {[Cu$_2$(6-Hmna)(6-mn)]·NH$_4$}$_n$ (1, 6-Hmna = 6-mercaptonicotinic acid, 6-mn = 6-mercaptonicotinate), was synthesized via the in situ cleavage of S–S bonds, using Cu(NO$_3$)$_2$·6H$_2$O, 6,6'-dithiodinicotinic acid (H$_2$dtdn) and pyrazine in a mixture of DMF and H$_2$O (5:1 v/v) under hydrothermal conditions at 140 °C (see Methods). Compound 1 was isolated in the form of brown, rectangular, rod-shaped crystals (Fig. 1; Supplementary Figs. 1 and 2). An FTIR spectrum of 1 showed a peak at 1697 cm$^{-1}$ which corresponds to the C=O stretching of an uncoordinated carboxylic group, three peaks at 1585, 1435, and 1312 cm$^{-1}$ corresponding to aromatic C=C stretching, and an additional peak at 525 cm$^{-1}$, corresponding to Cu–S bending (Supplementary Fig. 3).

The powder X-ray diffraction (PXRD) spectrum of 1 was in good agreement with the simulated pattern, thus confirming the purity of 1 (Fig. 2c). To confirm the structural integrity of the material, a PXRD pattern was recorded after annealing 1 at 100 °C, which confirmed that the crystallinity of compound 1 is well maintained even after annealing (Supplementary Fig. 4). A thermogravimetric analysis (TGA) showed that 1 is stable at temperatures up to 340 °C (Supplementary Fig. 5).

**Crystal structure.** A single-crystal X-ray diffraction analysis of 1 revealed that it crystallizes in the orthorhombic space group $Pna2_1$ (Supplementary Table 1). The asymmetric unit consists of two crystallographically independent Cu(I) centers, one 6-Hmna ligand, one 6-mn moiety, and one ammonium ion (Fig. 2a; Supplementary Fig. 6a). The Cu1 center is bound to three sulfur atoms (S1', S1, and S2) from two 6-Hmna ligands, one 6-mn

**Fig. 1** Synthesis of compound **1**. Synthetic pathway to {[Cu$_2$(6-Hmna)(6-mn)]·NH$_4$}$_n$

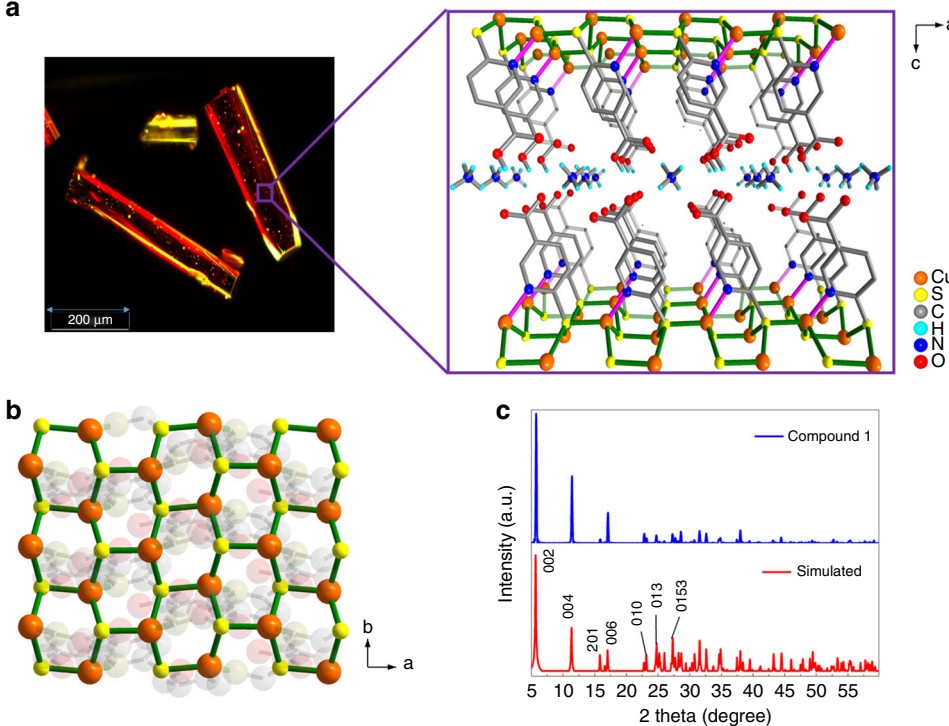

**Fig. 2** Characterization and structural properties of **1**. **a** Dark-field optical image and enlarged schematic view along the *c*-axis (Cu = orange, O = red, C = light gray, N = blue, S = yellow, H = cyan). **b** The corresponding 2D layer showing the Cu–S arrangement. **c** As-synthesized and simulated PXRD patterns

ligand, and one nitrogen atom (N1) from the other 6-mn ligand, resulting in a CuS$_3$N unit with a tetrahedral geometry. The coordination environment of the Cu2 center is similar to that of Cu1. Either the 6-Hmna ligand or the 6-mn moiety is coordinated to four Cu metal centers in a $\mu_4$-$\eta^1$-($\mu_3$-$\eta^1$:$\eta^1$:$\eta^1$) mode (Supplementary Fig. 6b, d). The Cu⋯Cu distances are in the range of 3.54–3.82 Å. The Cu–S bond lengths fall in the range of 2.303 (4)–2.384(3) Å and the Cu–N bond lengths are 2.006(10) and 2.061(10) Å. As shown in Fig. 2a, each {CuS$_3$N} unit cooperates with the two 6-Hmna ligands and two 6-mn ligands along the *b*-axis, forming vertical arrays that contain alternating arrangements of Cu and S atoms. Interestingly, these Cu and S atoms are arranged in a honeycomb-like structure in the *ab* plane (Fig. 2b). These (–Cu–S–)$_n$ sheets are arranged in a parallel manner and are separated by pendant arms from the 6-Hmna and 6-mn ligands, with a separation distance of 14.4 Å to form a layered MOF. In addition, ammonium ions (NH$_4^+$) are incorporated into the framework as counter ions, which are likely produced by the reduction of NO$_3^-$ in the presence of DMF and pyrazine under hydrothermal condition[29,30]. Hydrogen- bonding interactions between the ammonium ions and the neighboring carboxylic acid/carboxylate groups, appear to stabilize the framework (Supplementary Fig. 7). To further examine the structural topographies, BET analyses of compound **1** were carried out. The results indicate a low accessible surface area of 3.31 cm$^3$ gm$^{-1}$ at 758.4 torr (Supplementary Fig. 8), which defines it as a highly dense metal–organic framework.

Based on visualization with the naked-eye, the brown color of the crystals is indicative of an interesting oxidation state of Cu(I), which was corroborated by the summation of the bond valences (Supplementary Table 3). This conclusion was further confirmed by X-ray photoelectron spectroscopy (XPS) data. XPS peaks for 2$p_{3/2}$ and 2$p_{1/2}$ which appear at 931.5 and 951.3 eV, respectively, are characteristic peaks of the Cu(I) centers of **1** (Supplementary Figs. 9 and 11). XPS peaks for nitrogen observed at 398.4 eV

(N1*s*) and 401.1 eV (NH$_4^+$) confirmed the presence of nitrogen atoms in the structure (Supplementary Figs. 10 and 11)[31]. Interestingly, along with the above-mentioned structural features of compound **1**, these (–Cu–S–)$_n$ planes with unique arrangements provide an opportunity to examine the electron transfer characteristics of the material.

**Electrical conductivity measurement**. To investigate the electrical characteristics of the (–Cu–S–)$_n$ sheet, the temperature-dependent electrical conductivity of **1** was measured by the four-probe method for samples obtained from three different batches, with different single crystal thicknesses. The contacts for the electrical measurements were fabricated using a focused ion beam (FIB) method (Supplementary Fig. 12). The *I–V* curve was obtained for a single crystal by the four-probe method. The electrical conductivities for single crystals with thicknesses of 645, 957, and 1250 nm were 10.96, 4.18, and 1.81 S cm$^{-1}$, respectively (Fig. 3a). The small difference in the electrical conductivities for these crystals can be attributed to crystal defects[32], differences in contact resistance and the fact that the crystals were produced in different batches therefore, had different thicknesses. For MOF **1**, with a thickness of 645 nm, the contact resistance was calculated by the transmission line measurement (TLM) method. The contact resistance was determined to be 2550 Ω which was ~50% less than the total resistance (Fig. 3b)[33]. Temperature-dependent conductivity measurements for all of the crystals showed that the electrical conductivity increased with increasing temperature. As a result, it can be concluded that **1** shows semiconducting behavior (Fig. 3c). In addition, the activation energies for single crystals with thicknesses of 645, 957, and 1250 nm were determined to be 6, 6, and 9 meV, respectively. The low activation energy at low temperatures is indicative of electrical conduction through charge transfer via (–Cu–S–)$_n$ plane (Fig. 3d)[33]. The activation energies reported herein were calculated by the

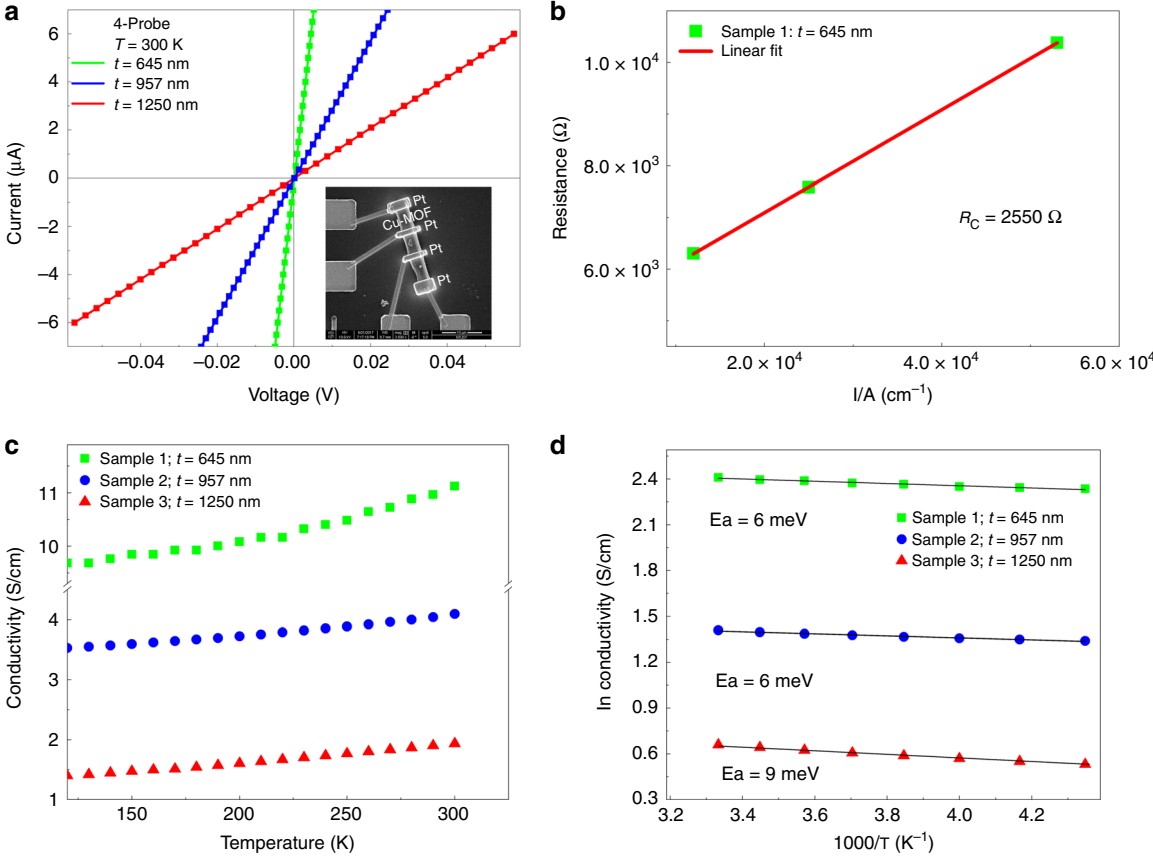

**Fig. 3** Electrical properties of **1**. **a** Current versus voltage curves for different thicknesses of **1** by four-probe measurement. **b** Contact resistance of sample **1** with a thickness of 645 nm. **c** Temperature-dependent conductivity of **1** at 0.1 V obtained by the four probe method. **d** Presentation of a relationship for the Arrhennius curve of **1** between ln (conductivity) and $1000/T$ ($K^{-1}$)

Arrhenius Eq. (1) as

$$\ln k = \ln A - \frac{E_a}{RT} \qquad (1)$$

where $k$ is the rate constant, $A$ is the pre-exponential factor, $E_a$ is the activation energy, $R$ is the universal gas constant, and $T$ is the absolute temperature in Kelvin[15].

It has been extensively confirmed in previous reports that the electrical conductivity of **1** would be enhanced through the $(-Cu-S-)_n$ chain within the framework through a bond approach[24–28]. For compound **1**, the major contribution to its high conduction value is the strong bond coupling between the $d$-orbitals of the copper(I) metal centers and the $p$-orbitals of the sulfur atoms, which not only contain $(-Cu-S-)_n$ chains but also contain a $(-Cu-S-)_n$ plane in the structure[34,35]. Thus, these $(-Cu-S-)_n$ sheets provide a suitable path for the transfer of electrons along the 2D plane, leading to a significant enhancement in electrical conductivity. This was corroborated by results of a DFT calculations (vide infra).

To the best of our knowledge, the electrical conductivity (10.96 S cm$^{-1}$) values are the highest that have ever been reported among MOFs in the form of a single crystal. After annealing sample **1** at 50 °C for 24 h, the electrical conductivity was determined to be 7.514 S cm$^{-1}$ and this value increased to 7.747 S cm$^{-1}$ at 50 °C, and then decreased slightly more to 7.508 S cm$^{-1}$ after cooling to room temperature (Supplementary Fig. 13). A single crystal of MOF **1** with a thickness of 645 nm was highly stable in air and this high electrical conductivity was maintained for periods of up to 18 h. The lowest measured electrical conductivity for sample **1** (7.51 S cm$^{-1}$) was comparable with its highest counterpart (10.96 S cm$^{-1}$). Although this substantial value is still in the same order of magnitude.

While some variations in morphology or topology from sample to sample may exist, the removal of water molecules that are adsorbed to the surface can cause corresponding changes in their electrical conductivities. Significantly, the values for electrical conductivity in repetitive measurements for individual single crystal samples are consistent, and essentially highly reliable (Supplementary Fig. 14).

**Bandgap investigations**. To investigate this issue further, the reflectance of this copper–sulfur-based MOF was measured using the diffuse reflection method, the results of which showed a broad peak centered at about 473 nm and a small peak at 868 nm (Supplementary Fig. 18, inset). The bandgap for the copper sulfur-based MOF was calculated by means of the Kubelka–Munk (K–M) function and the equation for which is given as

$$\frac{K}{S} = F(R) = \frac{(1-R)^2}{2R} \qquad (2)$$

where $K$ is the absorption coefficient, $S$ is a scattering factor, $R$ is the reflectance and $F(R)$ is the KM function. The indirect bandgap for the Cu MOF was estimated to be 1.34 eV by extrapolating a plot of $([F(R) \times h\upsilon])^{0.5}$ vs. $h\upsilon$) (Supplementary Fig. 18). Hence it is clear from the diffuse reflection that **1** is, in fact, a low bandgap semiconductor MOF, which is suitable for potential electronics applications[36]. It is unique among low bandgap MOFs, in that it is possible to significantly enhance the intrinsic electrical conductivity of the materials through a metal–sulfur plane.

**Electrochemical measurements**. The electrochemical redox behavior of **1** was investigated via the use of cyclic voltammetry (Supplementary Fig. 16). Cyclic voltammograms of the ligand, 6-mercaptonicotinic acid (6-Hmna), showed two irreversible peaks in the anodic region, at +0.5 and +1.5 V, respectively (Supplementary Fig. 17). In the cathodic region, two irreversible peaks observed were at −0.6 and −1.5 V. When the ligand is incorporated into the framework, similar peaks (+1.5, +0.5, −0.6, and −1.5 V) contributed by the ligands were also observed. At a negative potential, a broad peak is observed at ~−2.0 V. This reduction peak can be attributed to the reduction of Cu(I) to Cu (0). This Cu-based MOF (**1**) shows irreversible electrochemical behavior. The irreversible redox behavior of the ligand and the metal center suggests that the increased electrical conductivity of the copper–sulfur-based MOF **1** can be mainly attributed to the order inherent in a single crystal.

**Theoretical DFT study of Cu-MOF**. In order to gain deeper insights into the conduction behavior of **1**, first-principle DFT simulations were performed on its periodic structure (Supplementary Fig. 19) using the Vienna Ab-initio Simulation Package (VASP)[37] (details of the DFT simulations are given in the Methods section). The electronic band structure of **1** is shown in Fig. 4. The calculated band gap is determined by the energy difference between the valence band maximum (VBM) and the conduction band minimum (CBM). In the simulated band structure, the smallest bandgap is 1.203 eV which occurs at the $\Gamma$ (VBM) and $Z$ (CMB) points (as defined in the Brillouin zone of a primitive cell shown in Supplementary Fig. 20). It is highly important to note that most semiconducting MOFs typically have very flat band lines (i.e. bands with <0.05 eV dispersion widths) in their band structure due to weak orbital overlapping resulting from the weak hybridization of the orbitals of the constituent metal nodes and organic linkers. Surprisingly, **1** shows an anomalous band structure with a significantly steep dispersion, as shown in Fig. 4a. In particular, VBM is relatively flat along $X-\Gamma$ and $\Gamma-Y$ but suddenly decreases along the $\Gamma-Z$ or $Y-T$ directions. The energy differences between $\Gamma-Z$ and $Y-T$ in the VBM are 0.43 and 0.51 eV, respectively (see the expanded plot at the right in Fig. 4a). The values for these dispersion widths are high for any MOF with semiconducting behavior[36,38–42].

Importantly, this steep dispersion in the VBM is a clear sign that charge transport properties, including the conductivity of the material, are likely remarkable and extraordinary along the $\Gamma-Z$, as well as the $Y-T$ directions compared to other directions in this Cu-based MOF, which constitutes strong evidence that a higher conduction value arises in the direction along the $(-Cu-S-)_n$ plane. More specifically, as the dispersion diverges from a flat to a suddenly steep decrease, the charge effective mass turns into a lighter boosting the hole mobility, which increases the electrical conductivity.

Analysis of the density of states (DOSs) provides information regarding the origin of the bands arising from the elements (Fig. 4b); the blue line represents the total DOS while the other colored lines represent the partial density of the states (PDOS). The DOS shows that both the valence and conduction bands are composed of states of the Cu, C, N, O, S, and H atoms (Supplementary Fig. 21). The PDOS data reveal that the VBM largely consists of the states of Cu atoms, with some contributions

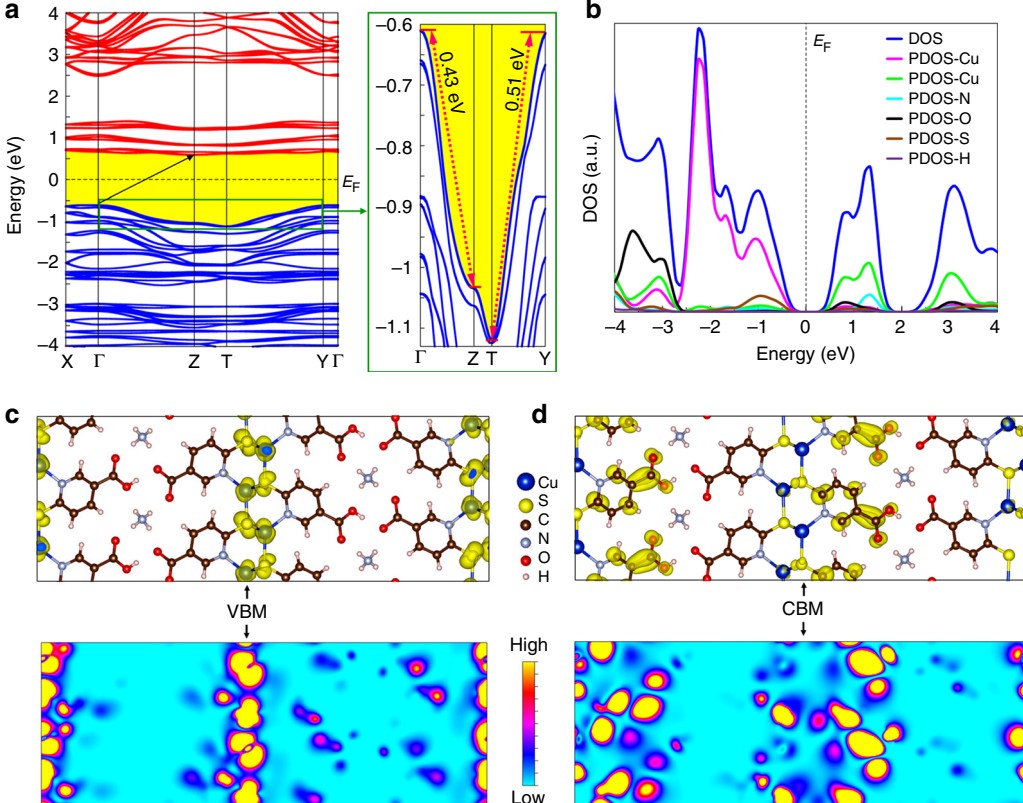

**Fig. 4** Electronic structure of **1**. **a** Band structure. On the right of the image, an expanded plot shows the steep dispersion of the valence bands. **b** DOS (blue line) and the PDOS (other colored lines) of **1**. The dashed black line at zero energy represents the Fermi level ($E_F$). Isosurfaces and contour plots of the band-decomposed partial charge density of **c** the valence band maximum (VBM) and **d** conduction band minimum (CBM). In combination with the PDOS, band-decomposed partial charge density plots show that the VBM is mainly composed of the states of Cu and S, and the $(-Cu-S-)_n$ plane generates a highly dense pathway for charge transport through it

from the states of S, and with nearly negligible contributions from states of the other elements in the ligand.

Importantly, the orbital-decomposed band structures clearly indicate that the VBM primarily originates from the hybridization of the 3d orbitals of copper and the 3p orbitals of sulfur (Supplementary Fig. 22). The CBM, on the other hand, is mainly composed of the states of C atoms, with small contributions from the O atoms, and minor states from Cu and S atoms. The contributions of H atoms to both the VBM and CBM around the Fermi energy level are negligible. In order to further clarify the features of the band edges, the band-decomposed partial charge density of the VBM and CBM are presented in Fig. 4c, d. Importantly, band-decomposed partial charge density plots show that the VBM is mainly composed of the states of Cu and S, and that the $(-Cu-S-)_n$ plane generates a highly dense pathway for charge transport through it (Fig. 4c). Significantly, our research findings show that **1** is, in fact, an indirect semiconductor with a small band gap of 1.203 eV, the band gaps of which are less than those of most previously reported semiconductor MOFs, thus making them suitable candidates for applications in the microelectronics industry[36,38–42].

In summary, a copper–sulfur-based MOF was successfully synthesized via the in situ cleavage of S–S bonds under hydrothermal conditions. These air stable crystals have the highest electrical conductivity (10.96 S cm$^{-1}$) among all single crystals of MOFs that have been reported to date. The high electrical conductivity originates from the integration of the $(-Cu-S-)_n$ plane in the structure. This is a new strategy for designing exceptionally high conductive MOFs. In addition, among its characteristic properties, the optical bandgap (1.34 eV) and theoretical bandgap (1.20 eV) permits this MOF to function as a low bandgap semiconductor with an electrical conductivity that increases with increasing temperature. Moreover, it has a low activation energy (6 meV), suggesting that a low energy would be required for charge transfer to occur. It is particularly noteworthy that this synthetic procedure is straightforward and simple, yet allows precise, self-assembly complexation to proceed. This significant finding offers a prototypical approach for producing a new class of highly electrically conductive metal–organic frameworks. We believe that these materials have substantial potential for use in the future. Further research is currently underway in our laboratory.

## Methods

**Preparation of {[Cu₂(6-Hmna)(6-mn)]·NH₄}ₙ (1)**. The copper-based metal–organic framework {[Cu₂(6-Hmna)(6-mn)]·NH₄}ₙ (**1**) was produced by the in situ cleavage of S–S bonds by reacting Cu(NO₃)₂·6H₂O (46.1 mg, 0.20 mmol), 6,6′-dithiodinicotinic acid (H₂dtdn, 30.8 mg, 0.10 mmol) and pyrazine (13.3 mg, 0.20 mmol) in a DMF/H₂O (5:1 v/v) solution at 140 °C for 72 h, under hydrothermal conditions followed by allowing the solution to slowly cool over a period of 72 h. The resulting brown rod-shaped crystals were isolated on a filter, washed with water, then DMF and, finally, allowed to dry at room temperature. Yield: 51.0% (15.9 mg, 0.035 mmol, based on the copper salt). IR data (KBr, cm$^{-1}$): 3163(w), 2980(m), 2847(m), 1685(m), 1582(s), 1435(s), 1385(w), 1366(w), 1334(m), 1254 (w), 1191(m), 1142(m), 1095(s), 1025(w), 847(m), 821(m), 772(s), 724(w), 529(m), 492(w). Anal. Calcd. for C₁₂H₁₁Cu₂N₃O₄S₂: C, 31.85; H, 2.45; N, 9.29; S, 14.17. Found: C, 31.34; H, 2.31; N, 9.10; S, 14.22.

**Electrical measurement and fabrication of single crystal**. The column-shaped micrometer-sized crystals of **1** were spread over insulating SiO₂ (300 nm)/n-Si templates that had been pre-patterned with a Ti/Au electrode before preparing the electrical contact. A Ti/Au electrode was used as an interconnection between the micrometer-sized crystals. The microelectrode was fabricated by a focused-ion beam (FIB) technique. Using the FIB technique, four platinum (Pt) electrodes were subsequently deposited on a column composed of crystals of **1**. The applied voltage was 30 kV and current was 100 pA for the ion beam decomposition of the Pt precursor in FIB-fabricated MOF devices, and post-annealing was not required to obtain an ohmic contact.

The lengths, widths, and thicknesses of the column-shaped crystals of **1** on the chip templates were estimated using field-emission scanning electron microscopy

(FESEM). An ultralow current leakage cryogenic probe station (LakeShore Cryotronics TTP4) was used for electrical characterization in the temperature range of 110–300 K. A semiconductor characterization system (Keithley 4200-SCS), with two independent source-measuring units was used as a source of the dc current/voltage and to measure the voltage/current. Electrical conductivity values for compound **1** are listed in Supplementary Table 2.

**Electrochemical measurements**. Cyclic voltammetry was carried out by means of a CH instrument 611B electrochemical analyser. The electrochemical behavior of **1** was investigated by using a three-electrode system, a glassy carbon electrode as the working electrode, Pt wire as the auxiliary electrode, and Ag/AgCl as the reference electrode with 0.1 M ([n-Bu₄N]PF₆)/ACN as a supporting electrolyte. Ferrocene was used as an internal standard. Before the CV measurements, a 0.1 M ([n-Bu₄N] PF₆)/ACN buffer solution (7 mL) was prepared and purged with a stream of nitrogen for 30 min. The Cu-based MOF was then ground into a powdered form (4.4 mg), which was mixed with 2 mL of acetonitrile and 25 μL of Nafion. The resulting slurry was used to prepare a thin layer on the glassy carbon electrode. A blank solution for cyclic voltammetry was measured and recorded in an acetonitrile potential window. The acetonitrile solvent was distilled over CaH₂ under nitrogen prior to use. The glassy carbon electrode was polished with alumina prior to each measurement. Oxygen was removed from the system by bubbling with N₂ gas for 20 min.

**X-ray photoelectron spectroscopy measurements**. XPS data for compound **1** was obtained using a PHI 5000 Versa Probe apparatus equipped with an Al Kα X-ray source (1486.6 eV). All measurements were done at room temperature.

**Details of DFT simulations**. All calculations were carried out using the periodic supercell arrangement with a plane-wave basis set, projector augmented-wave (PAW) potentials, and the Perdew–Burke–Ernzerhof (PBE) exchange-correlation functional. In addition, a Grimme's DFT-D3 correction was applied to account for dispersion forces (or van der Waals interactions). A plane-wave cutoff energy of 520 eV and Monkhorst-Pack 6 × 9 × 2 k-point meshes were used. The experimentally determined unit cell of **1** was optimized with the fixed lattice parameters at the experimental values until the maximum forces acting on each atom reached a value of <1 meV Å$^{-1}$ and the change in total energy was <10$^{-6}$ eV.

**PXRD characterization**. Experimental PXRD data were recorded on a Siemens D-5000 diffractometer at 40 kV, 40 mA for Cu Kα (λ = 1.5406 Å), with a step size of 0.02° and a scan speed of 0.52 s per step. Crystals of compound **1** was ground into a fine powder and placed on the silicon holder to collect PXRD data.

## Data availability

The X-ray crystallographic coordinates for structures reported in this study have been deposited at the Cambridge Crystallographic Data Centre (CCDC), under the deposition number 1814836. These data can be obtained free of charge from The Cambridge Crystallographic Data Centre via www.ccdc.cam.ac.uk/data_request/cif. The authors declare that the data which supports the findings of this study are available in the online version of the paper and its supplementary information file. All relevant data in support of the conclusions reached in this study can be obtained from the corresponding authors on request.

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

## Acknowledgements

We are grateful to the Academia Sinica and Ministry of Science and Technology, Taiwan for financial support. This work was also financially supported by the Center of Atomic Initiative for New Materials (AI-Mat), National Taiwan University, Taipei, Taiwan from the Featured Areas Research Center Program within the framework of the Higher Education Sprout Project by the Ministry of Education (MOE) in Taiwan.

## Author contributions

A.P. and M.U. prepared the draft of the manuscript with the input from all the authors. J.-W.S. and L.F.W. carried out the synthesis. T.-T.L. and S.M. resolved the crystal structure. A.P., Y.-S.C. and R.-S.C. carried out electrical measurement. B.S. and M.H. performed the DFT simulations. C.-M.N. carried out cyclic voltammetry. F.-R.C. reviewed the project. K.-L.L., K.-H.C., T.-W.T. and L.-C C. supervised the project.

## Additional information

**Competing interests:** The authors declare no competing interests.

