## [Peer Review File · Nature Communications]

Reviewers' comments:

Reviewer #1 (Remarks to the Author):

The manuscript "Integration of $(-Cu-S-)_n$ Plane and Redox Activity Towards Highly Conductive Metal-Organic Framework" by Chen, Tseng, Chen, and Lu et al. describes the synthesis and characterization of a metal-organic framework (MOF) based on copper and 6-mercaptionic acid. Although the described MOF has potentially interesting electronic conductivity, the authors fail to adequately demonstrate that the observed high conductivity is due to high structural order and redox activity. Additionally, the underlying structural integrity after annealing and lack of attention to detail in the electrochemical measurements necessitate major revisions both through further experiments and improvements to the writing. As such, this manuscript is not currently suitable for publication in Nature Communications and should be rejected in its current form. Further comments are outlined below, including both major and minor concerns for the consideration of the editor and benefit of the authors.

Major Concerns:

1. The manuscript suggests that the redox activity of both the ligand and the metal center play a role in the high conductivity of single crystals of 1. The authors attempt to use cyclic voltammetry to demonstrate this; however, their presentation of the electrochemical data and their electrochemical measurements section (page 14) is lacking in clarity and consistency. The authors note on page 14 that "the electrochemical behavior of 1 was investigated using a glass carbon electrode in a 1.0 M phosphate buffer solution containing acetonitrile...."; it is not clear what the exact electrochemical solvent mixture is, and this sentence appears to directly contradict the previous paragraph, which suggests that acetonitrile is used with an organic electrolyte. This is compounded by figure S11, which is labeled and referred to as a blank run but has multiple electrochemical features, suggesting the blank is not free of redox-active species and casting doubt upon the features observed for the ligand (figure S12) and 1 (figure 3a). Furthermore, there is no discussion about the other features observed in the cyclic voltammogram of 1 (figure 3a), including the different peak separations between the peaks attributed to the Cu redox couple and the ligand redox couple, the apparent differing numbers of electrons involved in the metal vs. ligand redox process, and the additional peaks present at ~ 1.3 V and -1.3 V. Taken together, these scans do not definitely establish that both the ligand and metal centers in their MOF are redox-active. Additionally, even if these electrochemical features are correctly assigned, the increased conductivity may instead be attributed to the order inherent to a single crystal, which the authors note is a rarity for conductive MOFs.
2. The single crystals of 1 used for conductivity measurements were investigated by thermogravimetric analysis (TGA), and revealed a mass loss of 4.6% which was attributed to H₂O loss; all crystals of 1 were then subjected to annealing at 50 °C. However, in the manuscript there is no discussion of structural integrity after this annealing procedure. This is concerning, as the H₂O molecules shown in the crystal structure (figure 1a) are in a hydrogen-bonding layer which "serve to stabilize the network", as the authors note on page 4. A structural rearrangement could be occurring, the structure could be collapsing, or the framework may now be porous. Given the authors' claims about conductivity in their crystals, this point must be more thoroughly corroborated.
3. The overall structure and writing in the manuscript needs to be reworked. At several points, the authors' make claims without adequately elaborating on or supporting them with evidence. For example, on page 8, they mention "the strong overlap occurred between the d-orbitals of the copper (I) metal centers and the p-orbital of the sulfur atoms to create the conductive path within the framework." However, this statement is made with no support in the text.

Minor Concerns

1. On page one, in the abstract the authors note that their single crystal MOF shows conductivity ~ 55 times higher than the most conductive MOF previously reported. However, they don't cite this example (which is presumed to be Yaghi et. al. Chem. Mater. 2012, 24(18), 3511).
2. On page 2, the authors state that poor orbital overlap between oxygen atoms and metal d-

orbitals is responsible for the poor conductivity of most MOFs; however, as noted in the reference above, the most conductive single crystal MOF is made up of metal-oxygen linkages, and this discrepancy is not resolved.

3. There is no investigation or discussion of the accessible surface area of 1, including after removal of the H₂O molecules at 50 °C.
4. On page 6, the authors mention that the activation energies are calculated by the Arrhenius equation (1). While this equation works for a classical semiconductor, Mott variable range hopping (VRH), which the authors claim their data supports, the exponent for VRH depends on the dimensionality of the conductor. VRH also operates in disordered systems, and so it may not apply at all in this system.
5. On page 7 figure c, their conductivity data stops at ~110 K, but their cryostat is reported on page 13 to be capable of reaching 20 K, and there is no discussion of why the authors did not measure conductivity at lower temperatures
6. On page S-11, figure S11 has the x-axis (potential) in an opposite order from the rest of the manuscript, creating unnecessary confusion.

Reviewer #2 (Remarks to the Author):

The manuscript by Lu and coworkers presents the preparation, the crystal structure, the electrical properties of a two-dimensional Metal-Organic Framework (MOF) based on the Cu(I)2S subunit. The compound shows a high conductivity, the highest for a MOF compound. Theoretical calculations confirm the experimental results on the semiconducting behavior and allow the authors to present a mechanism for the observed conduction involving the Cu-S units. Indeed the PDOS obtained from calculations show that the main pathway involves Cu(I) and sulfur highlighting the importance of the overlap between the Cu and the sulfur orbitals in promoting the conduction properties. Two interesting features are (1) the possibility to grow high quality single crystals and (2) the high chemical stability of the crystals in air.

The paper is well written even though some repetitions are present that can be expurgated. There are also some sentences whose meaning is not clear, for instance the following sentence: "The electrical conductivity of the copper sulfur-based MOF 1 can be attributed to both the bond approach and the redox behavior of the metal center and the organic linker (6-Hmna). Through the bond approach, the electrical conductivity of 1 could be enhanced from the [Cu2S]_n plane within the framework". It is not clear what the authors mean by "the bond approach"? There is one important issue that the authors may comment. The calculation of the band structure was made for the 3D compound. Would it be possible to perform calculations for only one 2D layer and compare the results with those of the 3D structure. This should allow extracting the contribution of the 2D Cu-S network and confirm the two-dimensional nature of the conductivity and eventually check how the 3D structure impacts the overall conductivity. This is important because it is well-known that the conductivity of one layer may be completely different from that of several layers with van der Waals interactions (direct versus indirect semi-conductor).

The search of highly conductive MOFs is a very active area of research that emerged only few years ago. I am surprised that the authors do not mention the work by Nishihara and coworkers (Univ. of Tokyo) who were among the first groups that highly contributed to this area, their work on the metal-sulfur containing 2D coordination network must be cited.

Computational details are in the Methods section and not in the Supporting Materials section as the authors state in the paper; this must be corrected.

I recommend acceptance of the papers providing the authors address the issue raised above.

Reviewer #3 (Remarks to the Author):

As requested, I will give comments only on the crystallographic part of the manuscript:

1) The structure analysis should be preferably performed in the standard setting of space group No. 33, which is Pna21 and not Pn21a as it is used in the manuscript (except there are good reasons given, not to do so).

2) There are obviously problems with the data set of the single crystal, which should be fixed: For many atoms the anisotropic displacement parameters were constrained to be equal (EADP) for the set of atoms N1 N2 C6 C7 C2 C10 C4 C9 and O1 O4 O2 O3 C1 C12, respectively.

This was probably done since a free refinement leads to physically meaningless non-positive definite ellipsoids or to very small ellipsoids with very different main axes.

In my opinion this points to a problem with the absorption correction or to a problem with the measurement itself (moving or mis-centered crystal for example).

If the original data set is available, there is in my opinion a good chance that the problem can be solved by a proper absorption correction (the correct parameters have to be given in SADABS). A free refinement of the displacement parameters should then result in physically meaningful ellipsoids for all atoms. If not, one has to think of other reasons for this behavior (non-merohedral twin for example?).

3) The reflections 020 and 0-20 (setting in Pn21a) should be removed from the refinement, as they are most likely partially shaded by the beamstop.

4) Special attention should be given to the hydrogen-atoms:

The displacement parameters were not refined but rather fixed at arbitrary values. For the final refinement they should be refined either free, or if this is not possible, via a constraint related to the displacement parameter of the atom to which it is attached (e.g. -1.2 for an "aromatic" H).

5) Another problem concerns the hydrogen atom positions attached the oxygen atoms.

It seems, that the water molecule is rather a H₃O⁺-ion, since there is a fairly large residual peak at the correct position for a third H-atom. Accordingly, I suspect that O3 is deprotonated and the current H-atom position at O3 is wrong.

6) Once the problems with the data set are fixed, please review carefully the positions and occupancies of the hydrogen atoms.

This is very important because it concerns the overall charge balance.

7) In general, the position of hydrogen atoms involved in the formation of hydrogen bonds should be refined freely if possible.

8) The scaling of the experimental powder diffraction pattern in Fig. 1c is such that the three strongest reflections are cut off. Please redo without cutting off the peaks and compare to the calculated pattern. If the intensities do not match, please explain (maybe preferred orientation because of anisotropic crystal shape?). Also a better resolution would be desirable in order to distinguish neighboring peaks and check purity.

Reviewer 1

1. The manuscript suggests that the redox activity of both the ligand and the metal center play a role in the high conductivity of single crystals of 1. The authors attempt to use cyclic voltammetry to demonstrate this; however, their presentation of the electrochemical data and their electrochemical measurements section (page 14) is lacking in clarity and consistency. The authors note on page 14 that “the electrochemical behavior of 1 was investigated using a glass carbon electrode in a 1.0 M phosphate buffer solution containing acetonitrile....”; it is not clear what the exact electrochemical solvent mixture is, and this sentence appears to directly contradict the previous paragraph, which suggests that acetonitrile is used with an organic electrolyte. This is compounded by figure S11, which is labeled and referred to as a blank run but has multiple electrochemical features, suggesting the blank is not free of redox-active species and casting doubt upon the features observed for the ligand (figure S12) and 1 (figure 3a). Furthermore, there is no discussion about the other features observed in the cyclic voltammogram of 1 (figure 3a), including the different peak separations between the peaks attributed to the Cu redox couple and the ligand redox couple, the apparent differing numbers of electrons involved in the metal vs. ligand redox process, and the additional peaks present at ~1.3 V and -1.3V. Taken together, these scans do not definitely establish that both the ligand and metal centers in their MOF are redox-active. Additionally, even if these electrochemical features are correctly assigned, the increased conductivity may instead be attributed to the order inherent to a single crystal, which the authors note is a rarity for conductive MOFs.

The electrochemical behavior of the material was reinvestigated under optimized conditions using a three-electrode system with Ag/AgCl as the reference electrode (Fig. S16 & S17). For the CV measurements, a buffer solution was prepared dissolving 0.1 M $[(n\text{-Bu}_4\text{N}]\text{PF}_6)/\text{ACN}$ (7 mL) followed by purging with a stream of nitrogen for 20 minutes. The Cu-based MOF (1) was finely ground into powdered form (4.4 mg) which was mixed with 2 mL of acetonitrile and 25 μL of Nafion. The resulting slurry was used to prepare a thin layer on the glassy carbon electrode. Details regarding the electrochemical solvent mixture can be found in the “Methods, Electrochemical Measurements” section (p15, L8-20). The cyclic voltammetry for a blank run has also been performed and the new cyclic voltammogram is included (Fig. S15; pS-12). Before the CV measurements, the oxygen was completely removed from the system by bubbling with N_2 gas for 20 min, hence, no additional peaks due to atmospheric oxygen were observed now in the blank run. The electrochemical measurements of ligand and compound 1 revealed that our Cu-based MOF (1) shows irreversible electrochemical behavior. After measuring the electrochemical behavior of compound 1 under improved conditions and performing advanced simulations, we agree with the reviewer that this high electrical conductivity of the Cu-based MOF (1) can mainly be attributed to the order inherent to a single crystal. Hence, the title and corresponding statements (p10, L7-17; p15, L8-20) have been revised in the manuscript to reflect the referee's suggestions. The new cyclic voltammograms are now included in Fig. S16 & S17 and Section “Electrochemical Measurements” has been rewritten with clarity and consistency (p10, L7-17).

2. The single crystals of 1 used for conductivity measurements were investigated by thermogravimetric analysis (TGA), and revealed a mass loss of 4.6% which was attributed to H_2O loss; all crystals of 1 were then subjected to annealing at 50 °C. However, in the manuscript there is no discussion of structural integrity after this annealing procedure. This is concerning, as the H_2O molecules shown in the crystal structure (figure 1a) are in a

hydrogen bonding layer which “serve to stabilize the network”, as the authors note on page 4. A structural rearrangement could be occurring, the structure could be collapsing, or the framework may now be porous. Given the authors’ claims about conductivity in their crystals, this point must be more thoroughly corroborated.

As per the suggestion made by reviewer 3, the structure of a new crystal of compound 1 has been resolved which confirms that there is no guest H₂O, instead NH₄⁺ cation is in the structure which makes the MOF stable up to 340 °C without any weight loss (Fig. S5; pS-7). The previously observed weight loss of 4.6% was due to the surface adsorbed water which were removed before re-measuring TGA by storing the crystals in a desiccator. PXRD was also performed for compound 1 before and after annealing at 100 °C (Fig. S4; pS-7) which confirms that the crystal structure is highly stable after annealing. In addition, hydrogen bonding

interactions are present between the NH_4^+ ions with carboxylic or carboxylate of the ligands which serve to stabilize the crystal structure as calculated by the platon software. Fig. S7; pS- 8 has been included and the corresponding statements on structure integrity have now been added to the text (p5, L18-20).

3. The overall structure and writing in the manuscript needs to be reworked. At several points, the authors' make claims without adequately elaborating on or supporting them with evidence. For example, on page 8, they mention "the strong overlap occurred between the d-orbitals of the copper (I) metal centers and the p-orbital of the sulfur atoms to create the conductive path within the framework." However, this statement is made with no support in the text.

After a careful proof-reading and revisions, the overall structure and writing of the manuscript has been significantly improved. Several references including new references (refs. 3-5, 20-22, 27-28, 32, 36-39) have been cited to support the statements in the text [(p2, L8, refs. 3-5); (p2, L12-14, ref. 10); (p2, L22-23, p3, L1-5, refs. 20-22); (p3, L8-9, ref.25); (p5, L16-18, ref. 27); (p6, L14-16, ref. 28); (p8, L2-5, ref. 32); (p11, L4-11, refs. 36-39)].

Minor Concerns

1. On page one, in the abstract the authors note that their single crystal MOF shows conductivity ~55 times higher than the most conductive MOF previously reported. However, they don't cite this example (which is presumed to be Yaghi et. al.Chem. Mater. 2012, 24(18),3511).

The recommended reference has been cited now as ref. 20 and the corresponding statement has been added [(p2, L22-23); (p3, L1)].

2. On page 2, the authors state that poor orbital overlap between oxygen atoms and metal d-orbitals is responsible for the poor conductivity of most MOFs; however, as noted in the reference above, the most conductive single crystal MOF is made up of metal-oxygen linkages, and this discrepancy is not resolved.

Most of the reported MOFs which exhibit conductive behavior are made up of metal-oxygen linkages. However, achieving a high value of electrical conductivity in MOFs has yet not been very successful without alleviating the energy mismatch in the metal-oxygen bonding as stated in ref. 25 (J. Am. Chem. Soc. 135, 8185-8188, (2013)). Recently, efforts have been successful in increasing the conductivity in a MOF by replacing the (M-O) bond with a (M-S) bond (refs. 23-26).

3. There is no investigation or discussion of the accessible surface area of 1, including after removal of the H_2O molecules at 50 °C.

As also suggested by reviewer 3, the crystal structure of compound 1 has been resolved and the results confirm that there is no guest H_2O , instead a NH_4^+ cation is in the structure which is very stable in the MOF. To investigate the accessible surface area, BET measurements of the sample after annealing were performed which showed a low surface area of $0.20 \text{ cm}^3/\text{gm}$ at 753.1 torr. The low surface area also indicates that the presence of NH_4^+ which is highly stable in the framework. Fig. S8; pS-9 has been included and corresponding statements have been added in the text (p5, L20-22).

4. On page 6, the authors mention that the activation energies are calculated by the Arrhenius equation (1). While this equation works for a classical semiconductor, Mott variable range hopping (VRH), which the authors claim their data supports, the exponent for VRH depends on the dimensionality of the conductor. VRH also operates in disordered systems, and so it may not apply at all in this system.

The activation energies were calculated by the Arrhenius equation (eq.1) which is a commonly known method used for conductive MOFs as reported in ref. 16. The corresponding statement on Mott variable range hopping (VRH) had been corrected as "...charge transfer via $(-Cu-S)_n$ plane...". The statements regarding variable range hopping have been revised in the text (p7, L18).

5. On page 7 figure c, their conductivity data stops at ~110 K, but their cryostat is reported on page 13 to be capable of reaching 20 K, and there is no discussion of why the authors did not measure conductivity at lower temperatures.

The cryogenic probe station (LakeShore Cryotronics TTP4) used for these measurements is capable to reach 20 K while using liquid helium as the cooling medium. However, mostly we used liquid nitrogen to conduct the measurements because the thermal activation behavior of majority carriers is more significant at the temperature range of 100-300 K than that at 20-100

K. Our samples show insensitive temperature dependence of conductivity, imply very low activation energies. In this case, it is not necessary to go for lower temperature down to 20 K. Hence, the conductivity of 1 was investigated from 110 to 300 K to demonstrate the semiconducting behavior of the single crystal. To avoid the potential confusion, the previous statement on the temperature limitation of the cryostat was corrected (p15, L4).

6. On page S-11, figure S11 has the x-axis (potential) in an opposite order from the rest of the manuscript, creating unnecessary confusion.

On page S-11, Fig. S11 (new Fig. S15; pS-12) has been revised with the correct order of the x-axis (potential) to reflect the referee's suggestion.

Reviewer 2

1. The manuscript by Lu and coworkers presents the preparation, the crystal structure, the electrical properties of a two dimensional Metal-Organic Framework (MOF) based on the $Cu(I)_2S$ subunit. The compound shows a high conductivity, the highest for a MOF compound. Theoretical calculations confirm the experimental results on the semiconducting behavior and allow the authors to present a mechanism for the observed conduction involving the Cu-S units. Indeed, the PDOS obtained from calculations show that the main pathway involves Cu(I) and sulfur highlighting the importance of the overlap between the Cu and the sulfur orbitals in promoting the conduction properties. Two interesting features are (1) the possibility to grow high quality single crystals and (2) the high chemical stability of the crystals in air. The paper is well written even though some repetitions are present that can be expurged.

There are also some sentences whose meaning is not clear, for instance the following sentence: "The electrical conductivity of the copper sulfur-based MOF 1 can be attributed to both the bond approach and the redox behavior of the metal center and the organic linker (6-Hmna).

Through the bond approach, the electrical conductivity of 1 could be enhanced from the $[\text{Cu}_2\text{S}]_n$ plane within the framework". It is not clear what the authors mean by "the bond approach"?

The bond approach has now been defined in the text (p3, L8-9) as "...bond approach, where the transfer of charge takes place via coordinate bonds." The term "the bond approach" has been used in the previous reported papers such as cited (ref. 25, J. Am. Chem. Soc. 2013, 135, 8185-8188).

2. There is one important issue that the authors may comment. The calculation of the band structure was made for the 3D compound. Would it be possible to perform calculations for only one 2D layer and compare the results with those of the 3D structure? This should allow extracting the contribution of the 2D Cu-S network and confirm the two dimensional nature of the conductivity and eventually check how the 3D structure impacts the overall conductivity. This is important because it is well-known that the conductivity of one layer may be completely different from that of several layers with van der Waals interactions (direct versus indirect semiconductor).

The calculation of the band structure was repeated for the 3D compound after crystal structure refinements. In addition, as per the suggestion made by the referee, the band structure of the 2D structure (Fig. A1) was also successfully calculated theoretically using the DFT calculation method with the same computational conditions as those for the 3D structure. A comparison of the band structures for the 2D and 3D configurations of 1 is given in Fig. A2. The calculated band gap of 2D structure is 1.286 which is wider (by 0.083 eV) than that of the 3D structure. For both 2D and 3D structures, the difference in the band gap values is relatively low and valence band features are similar. This is a reasonable result because VBM is dominantly originated from the states of the $(-\text{Cu}-\text{S}-)_n$ sheet which included in both structures. Meanwhile, shape of the conduction bands which mainly arise from the states of the ligands, is altered due to the fact that the ligands, are isolated in the 2D compared with that of 3D bulk. The conduction band minimum (CBM), on the other hand is similar to 3D one. Most importantly, these results for the band structures lead to the realization of how important the contribution of the 2D $(-\text{Cu}-\text{S}-)_n$ plane to the electronic properties of 1 actually is. Even when the 2D structure was extracted from the 3D bulk, the steep dispersion of VBM in the band structure is still preserved.

Fig. A1. 3D (a) and 2D (b) structures of 1. Both 3D and 2D structures contains $(-Cu-S-)_n$ sheet along ab -direction.

Fig. A2. Band structure of 3D (a) and 2D (b) structures of 1. VBM for both structures are similar due to the fact that they originated from the $(-Cu-S-)_n$ chain.

3. The search of highly conductive MOFs is a very active area of research that emerged only few years ago. I am surprised that the authors do not mention the work by Nishihara and coworkers (Univ. of Tokyo) who were among the first groups that highly contributed to this area, their work on the metal-sulfur containing 2D coordination network must be cited.

The work on metal sulfur containing 2D coordination network from Nishihara and coworkers is now cited in the main text as ref. 22 and the corresponding statement has been added (p3, L3-5) to reflect the reviewer's suggestion.

4. Computational details are in the Methods section and not in the Supporting Materials section as the authors state in the paper; this must be coorrected. I recommend acceptance of the papers providing the authors address the issue raised above.

The statement has been corrected in the text as "details of DFT simulations are stated in the Methods section" (p10, L21).

Reviewer 3

1. The structure analysis should be preferably performed in the standard setting of space group No. 33, which is Pna21 and not Pn21a as it is used in the manuscript (except there are good reasons given, not to do so).

We selected a new single crystal of compound 1 and reinvestigated its structure. The structure analysis was performed in the standard setting of space group Pna2₁. The newly refined crystal structure data are now listed in Table S1 and the CCDC (1814836) file has been updated. The corresponding statements and figures have been revised in the text (p5; L3; Section "Crystal Structure").

2. There are obviously problems with the data set of the single crystal, which should be fixed: For many atoms the anisotropic displacement parameters were constrained to be equal (EADP) for the set of atoms N1 N2 C6 C7 C2 C10 C4 C9 and O1 O4 O2 O3 C1 C12, respectively. This was probably done since a free refinement leads to physically meaningless non-positive definite ellipsoids or to very small ellipsoids with very different main axes. In my opinion this points to a problem with the absorption correction or to a problem with the measurement itself (moving or mis-centered crystal for example). If the original data set is available, there is in my opinion a good chance that the problem can be solved by a proper absorption correction (the correct parameters have to be given in SADABS). A free refinement of the displacement parameters should then result in physically meaningful ellipsoids for all atoms. If not, one has to think of other reasons for this behavior (non-merohedral twin for example?).

The problems with the data set have now been addressed. Interestingly a re-investigation of the crystal structure revealed the presence of ammonium ions in the structure, instead of guest water molecules. Due to the presence of an NH₄⁺ ion, a substantial effort was made to refine and solve the NDP issues. After this refinement, all of the data are able to meet the convergence and best operation results could be achieved. In addition, after replacing the O atoms with N atoms, the convergence results were found to be very stable and all the NDP errors could be eliminated.

3. The reflections 020 and 0-20 (setting in Pn21a) should be removed from the refinement, as they are most likely partially shaded by the beamstop.

The issue with the reflections 020 and 0-20 (setting in Pn21a) has been resolved. The space group has been changed to Pna21.

4. Special attention should be given to the hydrogen-atoms: The displacement parameters were not refined but rather fixed at arbitrary values. For the final refinement they should be refined either free, or if this is not possible, via a constraint related to the displacement parameter of the atom to which it is attached (e.g. -1.2 for an "aromatic" H).

The hydrogen-atoms were carefully refined in the revised crystal data where the ammonium ions are present in the structure, instead of guest water molecules.

5. Another problem concerns the hydrogen atom positions attached the oxygen atoms. It seems, that the water molecule is rather a H₃O⁺-ion, since there is a fairly large residual peak at the correct position for a third H-atom. Accordingly, I suspect that O3 is deprotonated and the current H-atom position at O3 is wrong.

The reinvestigation of the crystal structure of compound 1 revealed the presence of ammonium ions (NH₄⁺) in the structure, instead of guest water molecules or H₃O⁺ ions. Hence, the issue raised due to the hydrogen atom positions attached the oxygen atoms have been resolved. The new crystal refinement is quite good and very reasonable.

6. Once the problems with the data set are fixed, please review carefully the positions and occupancies of the hydrogen atoms. This is very important because it concerns the overall charge balance.

After re-calculating the crystal structure of compound 1, the position and occupancy of hydrogen atoms were carefully examined and the compound is now overall charge balanced.

7. In general, the position of hydrogen atoms involved in the formation of hydrogen bonds should be refined freely if possible.

The position of the hydrogen atoms involved in the hydrogen-bonding have also been refined freely.

8. The scaling of the experimental powder diffraction pattern in Fig. 1c is such that the three strongest reflections are cut off. Please redo without cutting off the peaks and compare to the calculated pattern. If the intensities do not match, please explain (maybe preferred orientation because of anisotropic crystal shape?). Also a better resolution would be desirable in order to distinguish neighboring peaks and check purity.

The scaling of the experimental powder diffraction pattern was repeated with complete reflections (without cut off) and the findings compared with the calculated pattern. The difference in intensities of some peaks as compared with the calculated pattern is due to the preferred orientation of the crystals. Resolution has been increased to distinguish neighboring peaks.

We are grateful to the reviewers for their constructive comments and valuable suggestions.

Reviewers' comments:

Reviewer #2 (Remarks to the Author):

The authors addressed all the issues raised in a complete manner. I, thus, recommend acceptance of the manuscript as is.

Reviewer #3 (Remarks to the Author):

Remarks concerning the crystallography:

Most of the problems mentioned in the previous review are resolved. The result was actually a new stoichiometry of the compound with ammonia as counter-ion instead of water.

However the following issues remain:

- 1) The crystal seems to suffer from extinction. As soon as the extinction parameter is refined one can see that according to the residual electron density the hydrogen atom at O3 should be located at O2, instead. This resolves also an alert in the checkcif-report. Actually, the situation might be a bit more complicated in reality, since both O-atoms seem to be disordered, which leads to disordered H-atoms, too. This may also be the reason for the unreasonable displacement parameters for O2. When refining the extinction parameter, no reflections have to be excluded from the refinement any more (as it has been done by the authors for 6 reflections).
- 2) Unfortunately, as a little flaw, the completeness of the data set is only 88% (any reason why?).
- 3) The indexing of the powder-XRD is wrong (Fig. 1 and Fig S4), it is still related to the non-standard setting of the space group as given in the first version of the manuscript. Also some reflection are most likely not correctly indexed: please keep in mind the single crystal was measured at low temperature, while the powder-XRD is probably recorded at room-temperature. This means the experimental pattern has lines which appear at slightly smaller diffraction angles compared to the calculated one (due to thermal expansion).
- 4) Please specify in the SI the instrumental parameters and the sample preparation for the powder-XRD

Even though initially I was asked to have a look on the crystallographic part only, I want to give some general remarks, too, because I think they are of relevance to the overall value of the manuscript:

- 1) In my opinion, calling the compound a "MOF" is at least a borderline case, if not inappropriate (see Batten et. al. Pure Appl. Chem., Vol. 85, No. 8, pp. 1715–1724, 2013), because: The layers containing the (Cu-S)_n - planes have no voids at all, and the potential voids containing the ammonium ions are not formed by a covalently bonded network, as you would expect for a MOF, but rather hydrogen bonds only. Removing the void-content (ammonium), will also necessarily destroy the structure. Actually, I would not classify this compound as a MOF at all.
- 2) The authors state: "Hence, we herein report on the preparation of single crystals of a Cu-based MOF consisting of a copper–sulfur plane, that shows an unprecedented high electrical conductivity (10.96 S/cm). This value is the highest electrical conductivity among MOFs in the form of a single crystal reported to date."

When having a look at table S2 and also literature, one can find several other coordination polymers/coordination networks with similar and even much higher conductivities, just not measured on a single crystal but rather on polycrystalline powders.

The authors should reformulate their statement such, that the impression is avoided that their compound is the MOF with the highest conductivity achieved to date.

- 3) I think, several aspects of the proposed chemistry behind the synthesis should be explained and backed up by corresponding citations:

In a typical MOF synthesis, DMF is often used as a mere solvent. I understand, that here the authors use DMF as a reducing agent not only to reductively cleave the S-S-Bond, but also to reduce Cu²⁺ to Cu¹⁺. Moreover, it seems that also the nitrate is reduced to ammonia (which I find surprising). Please explain and give references for similar observations as support for the

plausibility of such reductions by DMF.

Reviewer #4 (Remarks to the Author):

Lu and co-workers focus on an emergent area of research – conductive metal-organic frameworks. This field is pioneered by a number of groups including the Dinca, Yaghi, Allendorf, Long, or Kitagawa groups. My major concerns about the presented studies are the major claim about the highest conductivity in the “non-porous” MOFs and sufficient significance for the targeted journal. In addition, there are some issues with the literature citation/discussion. For instance, one of the main reviews related to this subject (Angew. Chem. Int. Ed. 2016, 55, 3566) published two years ago consists of the main table (page 3577) in which the conductivity values for the different MOFs include very high values 40 S/cm Ni₃(HITP)₂, 160 S/cm (Ni₃(BHT)₂), and others, which surpass the reported conductivity value in the presented studies (Unfortunately, the presented paper has not cited this review). A significant progress in the field has recently been made by introducing different concepts: guest inclusion (Allendorf and others), mixed-valanced metal sites MOFs (Dinca, Long, and others), nanowire insertion (Lin and others), sulfur/selenium linkers (Dinca and co-workers), dithiolene MOFs with M-S bonds in “dense” MOFs (Kambe et al). Therefore, there is a question – how these particular studies do shape the already existing landscape and why the presented work is suitable for the top journal? This particular question should be addressed in beginning of the manuscript since there are a number of studies in the literature, and some known MOFs surpass the presented conductivity value. The concept of metal-sulfur bond formation has been introduced before and led to very high conductivity value (e.g., 160 S/cm), and for instance, it has been also highlighted in the aforementioned review.

Other comments:

1. How do the authors support to the conclusion about the “indirect bandgap”? It is not clear from the text.
2. It is not clear from the text how the current CV data support the hypothesis from the Review 1 and why do authors agree with this suggestion.
3. The presented isotherm looks very unusual. I suggest to re-collect the data.
4. The authors claimed the presence of Cu⁺¹ according to the XPS data. Have any traces of Cu⁺² been detected which can also affect the conductivity values?
5. After incorporation of changes requested by the reviewers, the manuscript is not sufficiently polished. There are several word repetitions, and it is very difficult to read the presented manuscript with “new” additions.

Reviewers' comments:

Reviewer #2:

The authors addressed all the issues raised in a complete manner. I, thus, recommend acceptance of the manuscript as is.

Reviewer #3:

Remarks concerning the crystallography: Most of the problems mentioned in the previous review are resolved. The result was actually a new stoichiometry of the compound with ammonia as counter-ion instead of water. However, the following issues remain:

1) The crystal seems to suffer from extinction. As soon as the extinction parameter is refined one can see that according to the residual electron density the hydrogen atom at O3 should be located at O2, instead. This resolves also an alert in the checkcif-report. Actually, the situation might be a bit more complicated in reality, since both O-atoms seem to be disordered, which leads to disordered H-atoms, too. This may also be the reason for the unreasonable displacement parameters for O2. When refining the extinction parameter, no reflections have to be excluded from the refinement any more (as it has been done by the authors for 6 reflections).

Response:

Thanks for your valuable comments. Crystal structure refinement has been performed for the compound to resolve the remaining issue. After refining the extinction parameter, the hydrogen atom at O3 could be correctly relocated at O2 according to the residual electron density. No reflections were excluded from the refinement.

2) Unfortunately, as a little flaw, the completeness of the data set is only 88% (any reason why?).

Response:

After this new refinement, the completeness of the data set is now at 99.3% (to $\theta = 25.041^\circ$). It was re-refined, so that the crystal structure now has correct data files (including the Table, bond lengths and angles). The revised crystal refinement data has been updated at the Cambridge Structural Database (CCDC No. 1814836). Revised files (cif and checkcif) are attached.

3) The indexing of the powder-XRD is wrong (Fig. 1 and Fig S4), it is still related to the non-standard setting of the space group as given in the first version of the manuscript. Also some reflection are most likely not correctly indexed: please keep in mind the single crystal was measured at low temperature, while the powder-XRD is probably recorded at room-temperature. This means the experimental pattern has lines which appear at slightly smaller diffraction angles compared to the calculated one (due to thermal expansion).

Response:

The indexing of the XRD are now correctly labelled on the simulated pattern of the newly refined single crystal data (Fig. 1 and Fig. S4). Furthermore, the peaks in the measured patterns at room temperature closely match those in the simulated patterns that were calculated from crystal structure data.

4) Please specify in the SI the instrumental parameters and the sample preparation for the powder-XRD

Response:

The instrumental parameters and sample preparation for the powder XRD has now been included in the section "X-ray Diffraction Characterization" (Methods; P16, Para 3, L1-4).

Even though initially I was asked to have a look on the crystallographic part only, I want to give some general remarks, too, because I think they are of relevance to the overall value of the manuscript:

Additional comments

1) In my opinion, calling the compound a "MOF" is at least a borderline case, if not inappropriate (see Batten et. al. Pure Appl. Chem., Vol. 85, No. 8, pp. 1715–1724, 2013), because: The layers containing the (Cu-S)-n - planes have no voids at all, and the potential voids containing the ammonium ions are not formed by a covalently bonded network, as you would expect for a MOF, but rather hydrogen bonds only. Removing the void-content (ammonium), will also necessarily destroy the structure. Actually, I would not classify this compound as a MOF at all.

Response:

There are reports in the literature on the formation of layered MOFs that form 3D networks through hydrogen bonding. For example, Feng et.al. "A coronene-based semiconducting two-dimensional metal–organic framework with ferromagnetic behavior." *Nat. Commun.* **9**, 1, (2018) and Dan & Fang et.al. "Zinc-coordinated MOFs complexes regulated by hydrogen bonds: Synthesis, structure and luminescence study toward broadband white-light emission" *J. Solid State Chem.* **260**, 159 (2018). In our compound, the $(-\text{Cu}-\text{S}-)_n$ sheets are arranged in a parallel manner with a separation distance of 14.3 Å to form a layered MOF. Hence, this layered compound can be considered to be a borderline example of a MOF. The sentence "These $(-\text{Cu}-\text{S}-)_n$ sheets ... with a separation distance of 14.3 Å." is now revised to "These $(-\text{Cu}-\text{S}-)_n$ sheets ... with a separation distance of 14.3 Å to form a layered MOF." (P5, para 1, L13-15).

2) The authors state: "Hence, we herein report on the preparation of single crystals of a Cu-based MOF consisting of a copper–sulfur plane, that shows an unprecedented high electrical conductivity (10.96 S/cm). This value is the highest electrical conductivity among MOFs in the form of a single crystal reported to date. "When having a look at table S2 and also literature, one can find several other coordination polymers/coordination networks with similar and even much higher conductivities, just not measured on a single crystal but rather on polycrystalline powders. The authors should reformulate their statement such, that the impression is avoided that their compound is the MOF with the highest conductivity achieved to date.

Response:

The value for electrical conductivity is the highest among MOFs that have been measured in the form of a single crystal. As per the suggestions made by the reviewer, this statement has been reformulated as "Hence, we herein report on the preparation of single crystals of a Cu-based MOF consisting of a copper–sulfur plane, that shows an unprecedented high electrical conductivity (10.96 S/cm) measured on a single crystal." (P3, para 1, L16-18).

3) I think, several aspects of the proposed chemistry behind the synthesis should be explained and backed up by corresponding citations: In a typical MOF synthesis, DMF is often used as a mere solvent. I understand, that here the authors use DMF as a reducing agent not only to reductively cleave the S-S-Bond, but also to reduce Cu^{2+} to Cu^{1+} . Moreover, it seems that also the nitrate is reduced to ammonia (which I find surprising). Please explain and give references for similar observations as support for the plausibility of such reductions by DMF.

Response:

The reduction of nitrate to ammonium ions under hydrothermal condition using base and transition metal ions has been previously reported and these preparations are cited as ref. 30 (*Inorg. Chim. Acta.* **310**, 115, (2000)) and ref. 31 (*Coord. Chem. Rev.* **199**, 159, (2000)). In our case, the ammonium ions are likely produced by the reduction of NO_3^- in the presence of DMF and pyrazine under the hydrothermal conditions used. The statement "... ammonium ions (NH_4^+) are ... by the reduction of NO_3^- under hydrothermal ..." is now revised to "... ammonium ions (NH_4^+) are ...

by the reduction of NO_3^- in the presence of DMF and pyrazine under hydrothermal ...". (P5, para 1, L16-18).

Reviewer #4: Lu and co-workers focus on an emergent area of research – conductive metal-organic frameworks. This field is pioneered by a number of groups including the Dinca, Yaghi, Allendorf, Long, or Kitagawa groups. My major concerns about the presented studies are the major claim about the highest conductivity in the “non-porous” MOFs and sufficient significance for the targeted journal. In addition, there are some issues with the literature citation/discussion. For instance, one of the main reviews related to this subject (*Angew. Chem. Int. Ed.* 2016, 55, 3566) published two years ago consists of the main table (page 3577) in which the conductivity values for the different MOFs include very high values 40 S/cm $\text{Ni}_3(\text{HITP})_2$, 160 S/cm $(\text{Ni}_3(\text{BHT}))_2$, and others, which surpass the reported conductivity value in the presented studies (Unfortunately, the presented paper has not cited this review). A significant progress in the field has recently been made by introducing different concepts: guest inclusion (Allendorf and others), mixed-valanced metal sites MOFs (Dinca, Long, and others), nanowire insertion (Lin and others), sulfur/selenium linkers (Dinca and co-workers), dithiolene MOFs with M-S bonds in “dense” MOFs (Kambe et al). Therefore, there is a question – how these particular studies do shape the already existing landscape and why the presented work is suitable for the top journal? This particular question should be addressed in beginning of the manuscript since there are a number of studies in the literature, and some known MOFs surpass the presented conductivity value. The concept of metal-sulfur bond formation has been introduced before and led to very high conductivity value (e.g., 160 S/cm), and for instance, it has been also highlighted in the aforementioned review.

Response:

- (1). The review article (*Angew. Chem. Int. Ed.* **55**, 3566, (2016)) is now included and cited as ref. 29.
- (2). Several significant progress with different concepts are now discussed and are cited in the “Introduction” section {e.g. guest inclusion (Allendorf and others, ref. 16), mixed-valanced metal sites MOFs (Dinca, Long, and others, ref. 21), nanowire insertion (Lin and others, ref. 22), sulfur/selenium linkers (Dinca and co-workers, ref. 27), dithiolene MOFs with M–S bonds in “dense” MOFs (Kambe et al, ref. 23)}.
- (3). In our manuscript, a new concept is presented where the $(-\text{Cu}-\text{S}-)_n$ plane, instead of the $(-\text{Cu}-\text{S}-)_n$ chain is introduced in a MOF for the first time, resulting in a compound with a high electrical conductivity. This concept presents a significant advancement in the field of conducting metal–organic frameworks and this strategy provides a new model for designing highly conductive MOFs. The high electrical conductivities reported for some MOFs such as $\text{Ni}_3(\text{HITP})_2$ and $(\text{Ni}_3(\text{BHT}))_2$, were measured after they were formed into either pellets or thin films. It is noteworthy that our measurements of the electrical conductivity of compound were done on a single crystal (original form without any modifications) and this finding is significant, in that it is the highest among the MOFs where electrical conductivity was measured in the form of a single crystal.

Other comments:

1. How do the authors support to the conclusion about the “indirect bandgap”? It is not clear from the text.

Response:

In principle, the bandgap is referred to as an indirect bandgap, if the conduction band minimum (CBM) and the valence band maximum (VBM) occur at different points of the Brillouin zone in a material. For our Cu-MOF, as shown in Figure 3a, the CBM and VBM in the band structure were located at Z and Γ points of the Brillouin zone, showing the indirect bandgap. The zoomed-in view of Fig. 3a (Fig. R1) clearly displays that the indirect gap is 0.08 eV narrower than the direct one. This theoretical evidence indicates that the smallest band gap in the Cu-MOF is indirect. In addition, the experimental estimation by the extrapolating a plot of $([F(R) \times h\nu])^{0.5}$ vs $h\nu$ as presented in Figure S18 also shows that the Cu-MOF has an indirect bandgap.

Fig. R1. Electronic band structure of **1**.

2. It is not clear from the text how the current CV data support the hypothesis from the Review 1 and why do authors agree with this suggestion.

Response:

Cyclic voltammetry measurements of the ligand and compound **1** were carried out under optimized conditions. It was observed that the cyclic voltammograms of ligand and compound **1** display irreversible peaks. These CV data, in addition to the results of theoretical simulations, support the conclusion that the electrical conductivity can largely be attributed to the distinct $(-Cu-S-)_n$ plane of the single crystal, as suggested by reviewer 1.

3. The presented isotherm looks very unusual. I suggest to re-collect the data.

Response:

The BET data were recollected and the new isotherm (Fig. S8) is presented to replace the previous one.

4. The authors claimed the presence of Cu^{+1} according to the XPS data. Have any traces of Cu^{+2} been detected which can also affect the conductivity values?

Response:

The presence of Cu^{+2} ions was not detected after deconvolution of the XPS data (Fig. S9).

5. After incorporation of changes requested by the reviewers, the manuscript is not sufficiently polished. There are several word repetitions, and it is very difficult to read the presented manuscript with “new” additions.

Response:

After careful proof reading, the repetition of words in manuscript has been eliminated.

We are grateful to the reviewers for their constructive comments and valuable suggestions.

REVIEWERS' COMMENTS:

Reviewer #1 (Remarks to the Author):

The authors have addressed the issues raised and the paper presents enough new and interesting to be published in Nature Comm. I recommend acceptance as is.

Reviewer #3 (Remarks to the Author):

In my opinion the manuscript is of good quality, interesting, based on solid experimental work, it is concise and easy to read.

However, I have some concerns whether the results are really suited for this top journal: Indeed, the authors report the highest conductivity measured on a MOF single crystal. However, there are many MOFs/coordination polymers reported in literature, showing conductivities which are larger by orders of magnitude, just not measured on single crystals.

Comments:

In the abstract there is still the statement "unprecedented high electrical conductivity", without mentioning, that it refers only to measurements on single crystals.

The checkcif report gives still some avoidable alerts (level B):

why was the data set cut at 50 deg. 2Thata? The average I/sigma ratio is still about 12 at this diffraction angle and the Rint is still very good. Better do not cut the data and use also the reflections at higher angles. This will ensure a data/parameter ratio of at least 8 and will avoid the "Poor Data/Parameter Ratio" alert.

The other B-alert (Atom O2 has ADP max/min Ratio) might be a consequence of disorder, which is not well described, or it could be due to a problem with the absorption correction. Since all displacement parameters seem to be a bit small for this type of compounds, I suspect that the μ^*R value for the spherical absorption correction was too small (please check).

Reviewer #1:

The authors have addressed the issues raised and the paper presents enough new and interesting to be published in Nature Comm. I recommend acceptance as is.

Response:

Thank you very much!

Reviewer #3:

In my opinion the manuscript is of good quality, interesting, based on solid experimental work, it is concise and easy to read.

1) However, I have some concerns whether the results are really suited for this top journal: Indeed, the authors report the highest conductivity measured on a MOF single crystal. However, there are many MOFs/coordination polymers reported in literature, showing conductivities which are larger by orders of magnitude, just not measured on single crystals.

Response:

The two-sentence Editor's summary indeed highlights the importance of this manuscript. The summary is as follows. "Metal-organic frameworks that contain metal-sulfur chains have been demonstrated to exhibit good electrical conductivity. Here, the authors integrate a 2D metal-sulfur plane into a metal-organic framework, reporting a single crystal with a high conductivity of 10.96 S cm^{-1} ."

2) In the abstract there is still the statement "unprecedented high electrical conductivity", without mentioning, that it refers only to measurements on single crystals.

Response:

The statement “unprecedented” has been removed from the abstract as well as from the introduction.

3) Why was the data set cut at 50 deg. 2 Theta? The average I/σ ratio is still about 12 at this diffraction angle and the R_{int} is still very good. Better do not cut the data and use also the reflections at higher angles. This will ensure a data/parameter ratio of at least 8 and will avoid the “Poor Data/Parameter Ratio” alert.

Response:

As per the suggestion made by the referee, single crystal X-ray data were recollected at 2 theta > 50 degree. The new crystal data ensure a data/parameter ratio of 11 and avoid the “Poor Data/Parameter Ratio” alert.

4) The other B-alert (Atom O2 has ADP max/min Ratio) might be a consequence of disorder, which is not well described, or it could be due to a problem with the absorption correction. Since all displacement parameters seem to be a bit small for this type of compounds, I suspect that the μ^*R value for the spherical absorption correction was too small (please check).

Response:

After recollecting the single crystal X-ray data at 2 theta > 50 degree, the B-alert has now been removed.

We are grateful to the reviewers for their constructive comments and valuable suggestions.